# Understanding pathways to inequalities in child mental health: a counterfactual mediation analysis in two national birth cohorts in the UK and Denmark

Eric TC Lai [1,2] Daniela K Schlüter,[1,2] Theis Lange,[3,4] Viviane Straatmann,[1,5] Anne-Marie Nybo Andersen [2] Katrine Strandberg-Larsen,[2] David Taylor-Robinson[1,2]

[1]Institute of Population Health Sciences, University of Liverpool, Liverpool, UK
[2]Section of Epidemiology, Department of Public Health, University of Copenhagen, Copenhagen, Denmark
[3]Section of Biostatistics, Department of Public Health, University of Copenhagen, Copenhagen, Denmark
[4]Centre for Statistical Science, Peking University, Beijing, China
[5]Aging Research Center, Karolinska Institute & Stockholm University, Stockholm, Sweden

**Correspondence to**
Dr Eric TC Lai;
etcl@liverpool.ac.uk

## ABSTRACT

**Objectives** We assessed social inequalities in child mental health problems (MHPs) and how they are mediated by perinatal factors, childhood illness and maternal mental health in two national birth cohorts.

**Design** Longitudinal cohort study

**Setting** We used data from the UK Millennium Cohort Study and the Danish National Birth Cohort.

**Primary and secondary outcome measures** We applied causal mediation analysis to longitudinal cohort data. Socioeconomic conditions (SECs) at birth were measured by maternal education. Our outcome was child MHPs measured by the Strength and Difficulty Questionnaire at age 11. We estimated natural direct, indirect and total effects (TEs) of SECs on MHPs. We calculated the proportion mediated (PM) via three blocks of mediators—perinatal factors (smoking/alcohol use during pregnancy, birth weight and gestational age), childhood illness and maternal mental health.

**Results** At age 11 years, 9% of children in the UK and 3.8% in Denmark had MHPs. Compared with high SECs, children in low SECs had a higher risk of MHPs (relative risk (RR)=4.3, 95% CI 3.3 to 5.5 in the UK, n=13 112; and RR=6.2, 95% CI 4.9 to 7.8 in Denmark, n=35 764). In the UK, perinatal factors mediated 10.2% (95% CI 4.5 to 15.9) of the TE, and adding maternal mental health tripled the PM to 32.2% (95% CI 25.4 to 39.1). In Denmark, perinatal factors mediated 16.5% (95% CI 11.9 to 21.1) of the TE, and including maternal mental health increased the PM to 16.9% (95% CI 11.2 to 22.6). Adding childhood illness made little difference in either country.

**Conclusion** Social inequalities in child mental health are partially explained by perinatal factors in the UK and Denmark. Maternal mental health partially explained inequalities in the UK but not in Denmark.

## INTRODUCTION

Child and adolescent mental health problems (MHPs) constitute a substantial disease burden[1] affecting 10% to 20% of adolescents globally,[2] with around half of all lifetime cases of mental health disorders emerging by age 14.[3] Few studies have compared the social distribution and prevalence of MHPs across countries. One study from 2008 showed significant variation in MHPs across European countries, and on the basis of socioeconomic status, with the highest prevalence reported in the UK.[4] In the UK, according to the most recent longitudinal population-level data, child MHPs are increasingly common. One in eight children aged 10 to 15 years reported socioemotional behavioural problems from 2011 to 2012, compared with one in 10 in 2004.[5] According to some studies, Scandinavian countries like Denmark have also experienced an increase in incidence of child MHPs.[6]

There are clear social inequalities in child and adolescent MHPs on the basis of childhood socioeconomic conditions (SECs), as commonly measured by parental education, income or occupation. A systematic review of studies of the association between childhood SECs and child MHPs found that children growing up in disadvantaged childhood SECs were two to three times more likely to develop MHPs than their more advantaged peers, across studies in 23 countries.[7] Social inequalities in MHPs are evident early in life[8] and track strongly to adulthood.[9]

► We used two large contemporary cohorts in Europe with a wide range of information collected.
► Modern methods of causal mediation analysis were used to assess mediation by the putative mediators.
► One of the major limitations was that analysis could only be carried out in a harmonised manner across the two birth cohorts.
► As with most of the longitudinal cohort studies, missing data is inevitable and hence a challenge for analysis.

Few studies have assessed mediating pathways by which childhood SECs influence the risk of MHPs during late childhood/early adolescence.[7 10] There are many potential pathways, whereby children growing up in more disadvantaged SECs are more exposed or vulnerable to risk factors for subsequent MHPs. Studies have shown that infants born with low birth weight have a higher risk of MHPs in young adulthood,[11] and birth weight is highly socially patterned.[12 13] Moreover, maternal smoking during pregnancy, also a socially patterned risk factor, may be associated with higher risk of conduct problems in children.[14] Social disadvantage is associated with greater stress in parents and subsequent parental MHPs, impacting caregiving behaviours and quality.[15] In addition, risk factors intrinsic to the child such as chronic childhood illness are more common in children growing up in disadvantaged SECs, and may impact on subsequent risk of MHPs.[16]

Mäntymaa and colleagues categorise risk factors for child psychopathology as risks in the child, the parents and the social context.[17] Using this framework, we previously showed the importance of early years mediators in the UK, particularly perinatal factors, such as birth weight and gestational age, and family factors such as maternal MHPs.[10] Building on these findings, we aimed to compare causal pathways to inequalities in child MHPs in the UK and Denmark. We hypothesised that children growing up in more disadvantaged SECs are at increased risk of MHPs due to increased exposure to perinatal, maternal and child level risk factors. We further hypothesised that these pathways may differ across country contexts. In order to identify modifiable policy entry points to reduce inequalities in MHPs, we therefore compare pathways to MHPs in late childhood/early adolescence in two rich birth cohorts in the UK and Denmark.

## METHODS

### Study population

The Millennium Cohort Study (MCS) is a large nationally representative cohort of children born in the UK between September 2000 and January 2002 who have been followed up through six survey waves, when aged 9 months, and 3, 5, 7, 11 and 14 years.[18] The MCS initially recruited 19 244 families, of which 13 112 participated in follow-up at age 11. The Danish National Birth Cohort (DNBC) is a population-based cohort study. Between 1996 and 2002, 100 415 pregnant women, representing 30% of all pregnancies in Denmark during that period, were recruited at the first antenatal care visit with their general practitioner.[19] These pregnancies resulted in 96 853 live births out of which 35 764 participated in follow-up at age 11 (online supplemental figure 1).

### Exposure

Our primary exposure of interest was highest qualification attained by the mother at the time of their child's birth. This is a common measure of childhood SECs used in social epidemiological studies[20] and previous cross cohort comparisons of UK and Danish populations.[21] Details on how this measure was recorded are in the online supplemental appendix. We scaled the education measure in each country in order to derive the relative index of inequality in our models (RII).[22] The RII compares the risk of MHPs between children of highest and lowest SECs, taking into account the distribution of education level in the study population by ranking the maternal education groups from high to low and allocating a score (ranging from 0 to 1) that represents the midpoint of the category's range in the cumulative distribution (online supplemental appendix). We used this score as a continuous exposure variable in our regression model. The exponentiated coefficient gives a relative risk (RR), comparing children with highest and lowest SECs at birth.[22]

### Outcome

The main outcome of interest was MHPs measured at age 11, the longest follow-up that is currently captured in both cohorts, using the Strengths and Difficulties Questionnaire (SDQ) based on maternal report (online supplemental appendix). The SDQ has been shown to be a reliable screening instrument for emotional and behavioural problems in school-age children,[23] and has good internal consistency.[15] We used the well-established UK cut-offs for MCS, that is, 0 to 16 indicates normal to borderline behaviour and 17 to 40 indicates MHPs.[24] For DNBC, the cut-offs were: ≥17 for boys and ≥15 for girls indicating MHPs.[25]

### Potential mediators

In our previous study, we identified a range of childhood risk factors that potentially explain the social inequalities in adolescent mental health.[10] We mapped these potential mediators to those available at similar time points across both cohorts. These are shown in table 1, grouped into three categories: perinatal factors, childhood illness and maternal mental health.

### Covariates

Confounders were chosen on the basis of common causes of exposure (maternal education), mediators and outcome (MHPs at age 11 years).[26] Previous history of maternal MHPs and maternal age were considered to be confounders (online supplemental figure 2) (online supplemental appendix). We also adjusted for sex in our models.

### Statistical analysis

We undertook causal mediation analysis under the counterfactual framework to partition the total effect (TE) of maternal education on MHPs at age 11 years acting through the proposed mediators (natural indirect effect (NIE)) and through mechanisms that bypass the putative mediators (natural direct effect (NDE)) (online supplemental appendix).

**Table 1** Description of mediator variables

| Variables | Description DNBC | MCS |
|---|---|---|
| **Perinatal factors** | | |
| Smoking in pregnancy | Mothers were asked at, on average, 16 to 17 weeks of gestational age whether they smoked during pregnancy (yes/no) | Mothers were asked when the child was 9 months old whether they smoked before pregnancy (yes/no), and whether they changed after becoming pregnant (yes/no). Those who did not give up smoking during pregnancy were considered having smoked during pregnancy |
| Alcohol use in pregnancy | Mothers were asked at, on average, 16 to 17 weeks of gestational age about the number of units of alcoholic beverage namely beer, wine and spirit, that the mothers drank per week. Amount of alcohol consumption was categorised as: (1) did not drink alcohol during pregnancy; (2) light drinker: one to two units per week; (3) moderate drinker: three to six units per week; and (4) heavy drinker: seven units or more.[41] However, only a few observations falls into the category of heavy drinker (n<10); we collapsed the heavy drinker category into moderate drinker. A unit of alcohol was defined as one bottle of beer, one glass of wine, or one glass of spirits (about 4 cL), each of which corresponds to about 12 g of alcohol[42] | Mothers were asked when the child was 9 months old if they drank alcohol during pregnancy, the number of units they consumed per week. Amount of alcohol consumption was categorised as: (1) did not drink alcohol during pregnancy; (2) light drinker: one to two units per week; (3) moderate drinker: three to six units per week; and (4) heavy drinker: seven units or more.[41] The heavy drinker category was collapsed into moderate drinker category as some cells only had small number of observations (n~10) when cross-tabulated with maternal education. A unit of alcohol was defined as approximately half a pint of beer or one glass of wine, which is around 10 g of alcohol[43] |
| Birth weight | Data on birth weight in grams were obtained by data linkage to the Danish medical birth registry of Denmark | Mothers were asked when the child was 9 months old about the birth weight of their child, in kilograms or pounds. Birth weights were then converted to grams for analysis |
| Gestational age | Data on gestational age in days were obtained by data linkage to the Danish medical birth registry of Denmark | Gestational age in days was calculated on the basis of the mother's report of her expected due date[44] |
| Childhood illness at age 7 years | Mothers were asked whether the child had any handicap or chronic illness (yes/no) | Mothers were asked whether the child had any longstanding illness/disability/infirmity (yes/no) |
| Maternal mental health at age 7 years | Mothers were asked whether she had a psychiatric illness/bad nerve since birth (yes/no) | Maternal psychological distress was assessed using Kessler-6 scale,[45] asking whether in the last month how often respondents felt depressed, hopeless, restless or fidgety, worthless or that everything was an effort. Validated cut-off scores were used: normal (0 to 5); distress (6 to 24) |

DNBC, Danish National Birth Cohort; MCS, Millennium Cohort Study.

Our understanding of the temporal sequence of mediators[9] and the timing of measurement led us to choose a sequential approach to causal mediation analysis. We used logistic regression adjusted for maternal mental health before and during pregnancy and maternal age. We built three models (online supplemental appendix figure 1). Model 1 estimated the NIE through perinatal factors, including paths that operate through the downstream causal descendants of perinatal factors, but excluding the paths operating directly through childhood illness and/or maternal mental health at age 7 years. Model 2 estimated the NIE through both perinatal factors and childhood illness at age 7 years and their causal descendants but excluded the paths operating through maternal mental health at age 7 years. Model 3 estimated the NIE through perinatal factors, childhood illness and maternal mental health at age 7 years, encompassing all possible pathways but excluding the NDE from

maternal education to mental health at age 11 years. We estimated the RR and 95% CI for the NDE, NIE and TE sequentially, using the *medflex* package in R (V.3.5.1),[27] which parameterises the path-specific effects of interest in the presence of multiple mediators, taking into account potential interactions between the variables included in the mediating blocks.[27] We also estimated the proportion mediated (PM) in each model using the formula[28]:

$$\frac{\mathrm{RR_{NDE}}(\mathrm{RR_{NIE}}-1)}{(\mathrm{RR_{NDE}} \times \mathrm{RR_{NIE}}-1)}$$

The 95% CI for the PM were calculated using nonparametric bootstrapping for 1000 iterations. For the mediation analysis to have a causal interpretation, we assume no exposure-mediator interaction, that adjustment for confounding between exposure-mediator, mediator outcome and exposure outcome has been addressed, and that there is no post-treatment confounding.[28] In

both cohorts, we used multiple imputation to handle missing data (online supplemental appendix).

## Robustness tests

To test the robustness of our findings, we conducted sensitivity analyses. First, we repeated the analysis on the absolute risk scale using a linear probability model. The estimates derived from this model give the risk difference across extremes of the maternal education gradient (also interpretable as the slope index of inequality). Second, we checked for the presence exposure-mediator interaction by repeating Model 3, this time allowing for all two-way interactions between maternal education and the mediators in the model. We used a likelihood ratio test to examine if the model with interactions between maternal education and all mediators provided the better fit. Third, we undertook complete case analysis for those with complete observations for exposure, outcome, mediators and covariates. Fourth, we repeated the MCS analysis applying survey weights to account for sampling design and attrition (online supplemental appendix). Fifth, we repeated analyses using alternative measures of maternal mental health conditions at child age 7 years. The MCS questionnaire asked mothers whether they were ever diagnosed with depression or anxiety. We only considered the reported diagnoses after childbirth. In DNBC, we linked mothers' Det Centrale Personregister (CPR) number to hospital records via Statistics Denmark for any hospital contact for psychiatric illness since the child's birth. Finally, we conducted a bias analysis for unmeasured confounding, which assessed the sensitivity of the results to unmeasured confounding of the mediator-outcome association using Vanderweele's bias formula (online supplemental appendix).[29]

## Patient and public involvement

Patients and the public were not involved in this research.

## RESULTS
### Baseline characteristics

At age 11 years, 9% of children had MHPs in the UK, compared with 3.7% in Denmark (table 2). In both cohorts, mothers with lower education were more likely to be younger, have worse mental health, have smoked or consumed alcohol during pregnancy and have worse mental health when the child was 7 years old. Also, in both cohorts, children of mothers with lower education were more likely to have lower birth weight, shorter gestational age and longstanding illness at age 7 years (figure 1).

### Causal mediation analysis

In both cohorts, lower maternal education was associated with worse mental health at age 11. The TE of maternal education on MHPs (a RR comparing children with the highest and lowest SECs interpretable as the RII) for MCS children was 4.28 (95%CI 3.30 to 5.54) and for DNBC the TE was 6.21 (95%CI 4.94 to 7.80). In MCS, perinatal

**Table 2** Baseline characteristics of cohort participants in the UK MCS in wave 5 (age 11 years) and the DNBC at age 11 years

| MCS | |
|---|---|
| n (%) for categorical variables or mean (SD) for continuous variables | |
| Characteristics | |
| n | 13 112 |
| Maternal education | |
| Higher degree | 467 (3.8) |
| First degree | 1834 (14.9) |
| Diplomas in higher education | 1123 (9.1) |
| A/AS/S levels | 1266 (10.3) |
| GCSE grades A to C | 4197 (34.2) |
| GCSE grades D to G | 1285 (10.5) |
| None | 2103 (17.1) |
| Maternal mental health problem history | 3116 (24.7) |
| Boys | 6390 (50.5) |
| Maternal age (years) (SD) | 29.53 (5.9) |
| Socioemotional behavioural problem (SDQ score ≥17) | 1130 (9.0) |
| Birth weight (kg) (SD) | 3.37 (0.58) |
| Maternal smoking during pregnancy | 1867 (14.8) |
| Alcohol drinking during pregnancy | |
| Never | 11 505 (91.0) |
| 1 to 2 units per week | 557 (4.4) |
| ≥3 units per week | 580 (4.6) |
| Gestational age (days) (SD) | 276.20 (13.5) |
| Child's longstanding illness at age 7 years | 2195 (18.5) |
| Maternal mental health problem at child age 7 years (Kessler-6 score ≥6) | 2083 (18.9) |

| DNBC | |
|---|---|
| n (%) for categorical variables or mean (SD) for continuous variables | |
| Characteristics | |
| n | 35 764 |
| Maternal education | |
| Masters or above | 3851 (10.8) |
| Bachelor or equivalent | 10 789 (30.3) |
| Short cycle tertiary | 2074 (5.8) |
| Upper secondary | 15 948 (44.7) |
| Lower secondary or lower | 2992 (8.4) |
| Maternal mental health problem history | 2290 (6.7) |
| Boys | 17 920 (50.1) |
| Maternal age (years) (SD) | 30.36 (4.2) |

Continued

| Table 2 | Continued |
| --- | --- |
| **DNBC** | |
| Socioemotional behavioural problem (SDQ score ≥17) | 1375 (3.8) |
| Birth weight (kg) (SD) | 3.57 (0.6) |
| Maternal smoking during pregnancy | 7193 (20.9) |
| Alcohol drinking during pregnancy | |
| Never | 25 525 (74.2) |
| 1 to 2 units per week | 7875 (22.9) |
| ≥3 units per week | 980 (2.9) |
| Gestational age (days) (SD) | 279.15 (12.8) |
| Child's longstanding illness at age 7 years | 2114 (6.0) |
| Maternal mental health problem at child age 7 years (Kessler-6 score ≥6) | 4736 (13.6) |

DNBC, Danish National Birth Cohort ; GCSE, General Certificate of Secondary Education; MCS, Millennium Cohort Study; SDQ, Strengths and Difficulties Questionnaire.

factors mediated 10.17% of the TE (95% CI 4.47 to 15.87) (table 3). Adding childhood illness at 7 years in the model yielded little change to the PM (11.53%, 95% CI 5.20 to 17.86). However, adding maternal mental health at age 7 years almost tripled the PM (32.31%, 95% CI 25.37 to 39.06). In DNBC, perinatal factors mediated 16.47% of the TE (95% CI 11.88 to 21.06). As in the MCS, adding childhood illness at age 7 years did not substantially affect the PM (15.59%, 95% CI 9.86 to 21.31). Unlike in the

MCS, adding maternal mental health at age 7 years made little difference to the PM (16.91%, 95% CI 11.17% to 22.64%; $RR_{NIE}$ 1.16; 95% CI 1.10 to 1.23).

## Robustness tests

First, the analysis using the absolute risk scale showed a larger TE of maternal education on MHPs in the UK (10 percentage points, 95% CI 7 to 13) compared with Denmark (6 percentage points, 95% CI 5 to 7) (online supplemental table 1). The pattern of mediation on the absolute scale was similar to that on the relative scale. Second, models with all two-way interaction terms did not have a better fit in either cohort (likelihood ratio test p value >0.05). Third, repeating the analysis using complete cases showed similar patterns of mediation as in the main analysis (online supplemental table 2). Fourth, applying survey weights in the MCS data also yielded similar patterns of mediation, though estimates were slightly attenuated (online supplemental table 3). Fifth, we used alternative measures of maternal mental health at age 7 years in both cohorts. In MCS, using maternal reported 'ever diagnosis of depression' led to a reduced PM in Model 3 (11.19%, 95%CI 4.82 to 17.56). In DNBC, using any psychiatric diagnosis as captured in the population registry did not alter the results (online supplemental table 4). Finally, the bias analysis showed that the estimated NIEs were robust to the presence of moderate unmeasured confounding (online supplemental table 5).

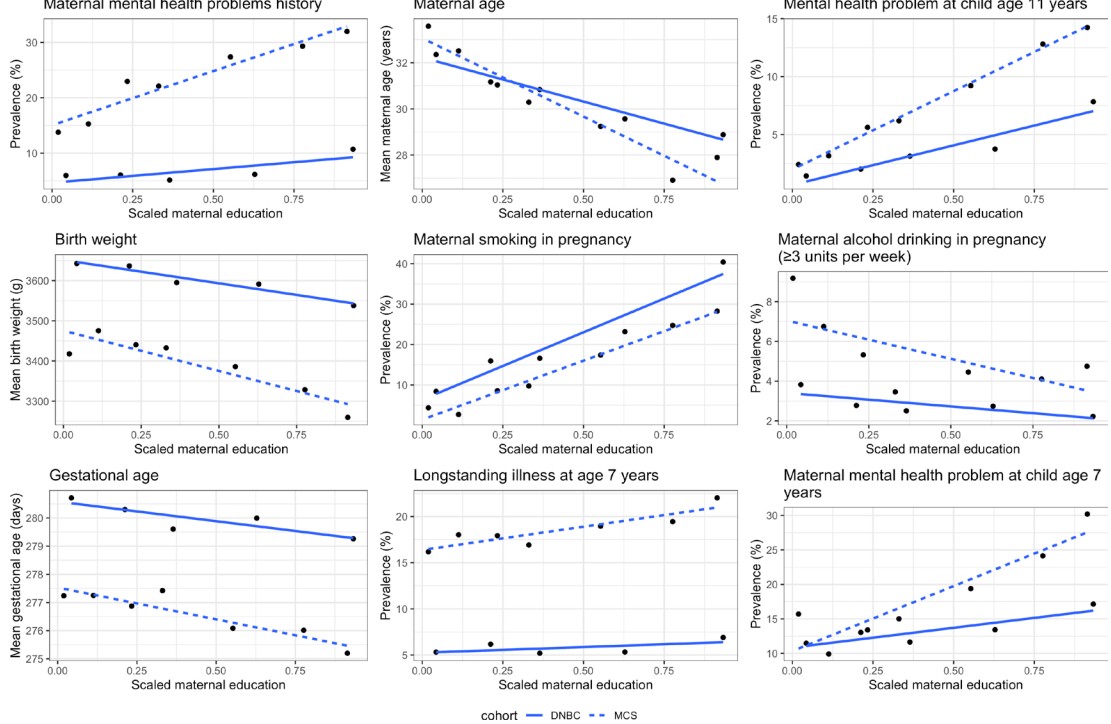

**Figure 1** Socioeconomic gradient of baseline characteristics in the UK MCS and the DNBC. DNBC, Danish National Birth Cohort; MCS, Millennium Cohort Study.

**Table 3** Estimates from causal mediation analysis for the association of maternal education and socioemotional behavioural problems at age 11 years in the UK MCS and the DNBC

| Mediator | Effect | RR | 95% CI | PM | 95% CI |
|---|---|---|---|---|---|
| **MCS** | | | | | |
| Perinatal factors | Natural direct effect | 3.95 | 3.05 to 5.12 | 10.17 | 4.47 to 15.87 |
| Perinatal factors | Natural indirect effect | 1.08 | 1.03 to 1.14 | . | . |
| Perinatal factors | Total effect | 4.28 | 3.30 to 5.54 | . | . |
| +Childhood illness at age 7 years | Natural direct effect | 3.89 | 3.01 to 5.04 | 11.53 | 5.20 to 17.86 |
| +Childhood illness at age 7 years | Natural indirect effect | 1.09 | 1.04 to 1.16 | . | . |
| +Childhood illness at age 7 years | Total effect | 4.28 | 3.30 to 5.54 | . | . |
| +Maternal mental health at age 7 years | Natural direct effect | 3.15 | 2.44 to 4.06 | 32.21 | 25.37 to 39.06 |
| +Maternal mental health at age 7 years | Natural indirect effect | 1.33 | 1.23 to 1.44 | . | . |
| +Maternal mental health at age 7 years | Total effect | 4.28 | 3.30 to 5.54 | . | . |
| **DNBC** | | | | | |
| Perinatal factors | Natural direct effect | 5.26 | 4.16 to 6.64 | 16.47 | 11.88 to 21.06 |
| Perinatal factors | Natural indirect effect | 1.16 | 1.11 to 1.21 | . | . |
| Perinatal factors | Total effect | 6.21 | 4.94 to 7.80 | . | . |
| +Childhood illness at age 7 years | Natural direct effect | 5.25 | 4.17 to 6.61 | 15.59 | 9.86 to 21.31 |
| +Childhood illness at age 7 years | Natural indirect effect | 1.15 | 1.09 to 1.21 | . | . |
| +Childhood illness at age 7 years | Total effect | 6.21 | 4.94 to 7.80 | . | . |
| +Maternal mental health at age 7 years | Natural direct effect | 5.19 | 4.12 to 6.53 | 16.91 | 11.17 to 22.64 |
| +Maternal mental health at age 7 years | Natural indirect effect | 1.16 | 1.10 to 1.23 | . | . |
| +Maternal mental health at age 7 years | Total effect | 6.21 | 4.94 to 7.80 | . | . |

.DNBC, Danish National Birth Cohort; MCS, Millennium Cohort Study; PM, proportion mediated; RR, relative risk.

## DISCUSSION

Using national birth cohort data from two countries, our study shows that children in the UK have higher prevalence of MHPs at age 11 years compared with children in Denmark. Relative inequalities were stark in both countries, with roughly four times and six times higher risk for children at bottom of SECs scale compared with the top, in the UK and Denmark, respectively. Absolute inequalities were larger in the UK. Perinatal factors explained 10% of the social inequality in the UK and 16% in Denmark. By contrast, maternal mental health was an important mediator only in the UK in our primary analysis, with the final model explaining 32% of the relative inequality at age 11 years.

### Comparison with other studies

In this study, we found that in the UK sample, by age 11 years, around 9% of children had MHPs, whereas in Denmark the figure was around 3.7%. These findings broadly correspond with recent findings that one in 10 UK children aged 5 to 15 years of age has MHPs.[30] While we lack contemporary comparative data on child mental health, our findings corroborate the Kidscreen study from 2008, which examined 15 945 adolescents across 13 European countries using adolescent self-reported SDQ. The authors found that the UK had the worst adolescent mental health, with the largest effect size for the association of low SECs on SDQ score. The study did not include Denmark, but the low prevalence of MHPs in DNBC are comparable to findings for Germany (2.9%) and Switzerland (3.6%).[4]

Our results show clear social inequalities in MHPs in adolescents in both the UK and Denmark. This finding is corroborated by numerous previous studies, which showed a socioeconomic gradient of MHPs in children/adolescents in various settings.[7] Relative inequalities were similar in both countries, with a greater point estimate in Denmark. However, relative inequalities can increase when the overall prevalence in the population is low,[31] as is the case for MHPs in Denmark. On the absolute scale, inequalities were larger in the UK, with a 10 percentage point difference across the maternal education hierarchy, compared with 6 percentage points in Denmark.

Our study showed that perinatal factors (smoking and alcohol use in pregnancy, gestational age and birth weight) explained around 10% of the socioeconomic gradient of MHPs in late childhood/early adolescence. Both maternal smoking and alcohol use during pregnancy are socially patterned and are associated with children's subsequent risk of conduct problems in childhood and adolescence, with some evidence that these associations may be causal.[14 32] A substantial proportion of women smoke during pregnancy in the UK and Denmark[33] with

smoking more prevalent among socially disadvantaged women.[34 35] There are also clear inequalities in low birth weight and preterm delivery, which are associated with increased risk for childhood MHPs, potentially as a result of insults to early brain development.[9]

Children with chronic physical illnesses have greater vulnerability to psychosocial problems: they usually have less perceived control over the progression of the relevant disease, and are more anxious about symptom onset, peer rejection and the restriction of daily activities.[36] However, in our analysis, adding childhood illness at age 7 years did not explain a substantial difference in inequalities, over and above those explained by our perinatal risk factor variables. A possible explanation is that causal pathways from childhood illness to MHPs at age 11 years might have descended from perinatal factors. It is also possible that previous evidence of the association of childhood illness and MHPs might have reflected underlying unadjusted confounding by SECs.

Maternal mental health measured up to age 7 years appeared to be an important mediator in the UK, but not Denmark. Maternal mental health is a well-established risk factor for child MHPs and has been identified as a mediator of the association between SECs and child mental health outcomes in a number of previous studies.[15] In another UK birth cohort, ALSPAC (Avon Longitudinal Study of Parents and Children), MHPs showed an intergenerational pattern, that is, poor mental health could be transmitted from mothers to children.[37] The lack of mediation by maternal mental health in DNBC could reflect true underlying differences between contexts, and the observed social gradient in maternal MHPs is much shallower in Denmark compared with the UK (figure 1). However, it is possible that the Kessler-6 scale used in the MCS and the ever-diagnosed psychiatric illness question used in DNBC capture different constructs. The Kessler-6 scale captures maternal mental health in the 30 days prior to the questionnaire being administered, whereas the ever-diagnosed question used in MCS captures any psychiatric illness history experienced by the mother. The Kessler-6 scale also captures other dimensions of mental health other than feeling depressed, including hopelessness, restlessness, fidgety, worthlessness and whether everything was an effort. Repeating the MCS analysis with an alternative measure of maternal reported mental reduced the PM. Future studies with more comparable mediator data could explore this finding further.

## Strengths and limitations

One of the key strengths of our study is the use of two large contemporary cohorts in Europe. A wide range of information was collected in these cohorts, allowing harmonisation of variables of interest, and an examination of whether mediating mechanisms were consistent across settings. Also, as suggested by Goodman and colleagues, we also applied country-specific cut-offs for SDQ total difficulty scores to improve the validity of cross-country comparisons.[38]

However, this study also has some limitations. First, as outlined above with regard to maternal mental health, differences in, and availability of, variables in our respective cohorts, limited the extent to which we could explore potential mediating pathways in a harmonised manner across both cohorts. For example, it is plausible that childhood SECs might influence mental health outcomes in late childhood/early adolescence via quality of family relationships or parenting style, which is measured in the MCS but not in the DNBC. Furthermore, data about potentially mediating childhood adversities such as domestic violence, sexual abuse and parental criminality were not available for inclusion in our analysis. Second, although we used modern methods for causal mediation analysis, and adjusted for a range of potential confounders, the assumption of complete adjustment of confounding is still required for causal interpretation of our estimates (online supplemental appendix). However, our bias analysis showed that our results are robust to presence of unmeasured confounding of moderate strength. Third, missing data is a limitation for many longitudinal studies. Nevertheless, sensitivity analysis comparing imputed and complete case analyses showed similar results. Finally, it is possible that children with mentally distressed mothers might report their children's mental health more negatively.[39] As a consequence, the measured indirect effect for maternal mental health might have been inflated in this study.[40]

## Implications for policy

The risk of child MHPs is much greater for disadvantaged children in both the UK and Denmark. Our findings suggest that public health programmes address perinatal risk factors and that support to optimal maternal mental health may reduce inequalities in child MHPs. In addition, given the unexplained residual inequality, to reduce MHPs in childhood, policy action is needed to address the upstream determinants of child mental health, with a focus on reducing socioeconomic inequalities.

**Acknowledgements** The authors thank the families who participated in the follow-up of the Millennium Cohort Study and the Danish National Birth Cohort and provided the valuable data. The Danish National Birth Cohort was established with a significant grant from the Danish National Research Foundation. Additional support was obtained from the Danish Regional Committees, the Pharmacy Foundation, the Egmont Foundation, the March of Dimes Birth Defects Foundation, the Health Foundation and other minor grants. The DNBC Biobank has been supported by the Novo Nordisk Foundation and the Lundbeck Foundation. Follow-up of mothers and children have been supported by the Danish Medical Research Council (SSVF 0646, 271-08-0839/06-066023, 0602-01042B, 0602-02738B), the Lundbeck Foundation (195/04, R100-A9193), The Innovation Fund Denmark 0603-00294B (09–067124), the Nordea Foundation (02-2013-2014), Aarhus Ideas (AU R9-A959-13-S804), University of Copenhagen Strategic Grant (IFSV 2012) and the Danish Council for Independent Research (DFF - 4183-00594 and DFF - 4183-00152).

**Contributors** ETCL did the statistical analysis and wrote the first draft with input from DT-R. ETCL, DT-R and KS-L conceptualised the study. DKS and TL helped design statistical analysis. DKS, VS, AMNA and KS-L contributed to data interpretation. All authors have critically reviewed and approved the final version of the manuscript for publication.

**Funding** ETCL and DTR are funded by the MRC on a Clinician Scientist Fellowship (MR/P008577/1). DKS was funded by the Cystic Fibrosis Trust (CF EpiNet SRC). TL, KS-L and A-MNA are funded by the University of Copenhagen. The funders had no

role in study design, data collection and analysis, decision to publish or preparation of the manuscript.

**Competing interests**  None declared.

**Patient and public involvement**  Patients and/or the public were not involved in the design, or conduct, or reporting, or dissemination plans of this research.

**Patient consent for publication**  Not required.

**Ethics approval**  The MCS was reviewed and approved by appropriate research ethics committees at each wave of data collection, and parents provided written informed consent for all components of the MCS. All DNBC participants provided written consent and ethical approval was obtained from the Danish Data Protection Agency (11-year follow-up approval number: 2009-41-3339). The current study was approved by the DNBC management and Steering Committee.

**Provenance and peer review**  Not commissioned; externally peer-reviewed.

**Data availability statement**  Data may be obtained from a third party and are not publicly available. The data of the Millennium Cohort Study could be obtained from the UK Data Service; and the data for Danish National Birth Cohort could be obtained upon application filed to Statistics Denmark and subject to approval. Computing code could be obtained by emailing the corresponding author (ETCL).

**ORCID iDs**
Eric TC Lai http://orcid.org/0000-0002-1229-9471
Anne-Marie Nybo Andersen http://orcid.org/0000-0002-4296-8488

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
