## [Reviewer comments · BMJ Open]

ARTICLE DETAILS

TITLE (PROVISIONAL)	Understanding pathways to inequalities in child mental health: A counterfactual mediation analysis in two national birth cohorts in the UK and Denmark
AUTHORS	Lai, Eric TC; Schlüter, Daniela K; Lange, Theis; Straatmann, Vivianne; Andersen, Anne-Marie Nybo; Strandberg-Larsen, Katrine; Taylor-Robinson, David

VERSION 1 – REVIEW

REVIEWER	Lucy Thompson University of Glasgow, UK
REVIEW RETURNED	18-Jun-2020

GENERAL COMMENTS	Congratulations on a very good quality paper assessing an important question using good quality cohort data in two different countries. You have acknowledged the limitations of the datasets, especially in terms of differential attrition, and have taken steps to mitigate these inasmuch as is possible. The stark difference between the UK and Denmark is somewhat depressing and reflects the social differences between the two nations - a paper like this hopefully contributes to the relevant socio-political agendas. I am not a statistical specialist, so I don't feel I can comment on the analysis in any detail. However, as a non-specialist I was able to follow what you had done and why, and found the interpretation to be appropriate. A small point - there was one sentence that I didn't think quite made sense. On p23 (line 29-33) you say 'Finally, it is possible that children with psychologically distressed mothers might report their children's mental health more negatively' - would it not be better to say 'Finally, it is possible that mothers who are mentally distressed might report their children's mental health more negatively'? Overall, an excellent paper - I I
--

REVIEWER	Apolinaras Zaborskis Lithuanian University of Health Sciences, Lithuania
REVIEW RETURNED	24-Jun-2020

GENERAL COMMENTS	Many thanks for giving me the opportunity to review this manuscript entitled "Pathways to inequalities in child mental health – Evidence from two national birth cohorts in the UK and Denmark". Using national birth cohort data from two countries (UK and Denmark), the authors analyzed the pathways of social inequalities in child mental health problems (MHP) and how they are mediated by perinatal factors, childhood illness and maternal mental health.
---

	The authors hypothesised that children growing up in more disadvantaged socio-economic conditions (e.g. with lower mother education) are at increased risk of MHPs due to increased exposure to perinatal, maternal and child level risk factors. They also hypothesised that these pathways may differ between countries. The findings from this study may be useful to readers who interested not only in children's mental health but also in statistical analysis when it is necessary to assess the complex effects of mediator factors on outcomes. The article is written in a typical format. The overall rationale, design and objectives of the data analysis conducted in this study are presented clearly and concisely. There is an excellent description of the study design, sampling frame and data collection methodology for the dataset for this study. Statistical analysis was conducted using modern techniques and methodologies. A detailed supplementary appendix explains the calculations performed. Since I am not a researcher in this field, I have no substantive comments on the content of this article, except some minor points that might be taken in account for revision of the manuscript or can be considered just as questions.  1. There are many tables in the article, but the data in them have little comment. This is especially true for supplementary tables. 2. Table 3: How was estimated PM? Clearly, the formula presented on page 13 is not appropriate for this assessment. 3. Is it appropriate to use path analysis for the causal mediation analysis? Thank you for considering my opinion.
--	--

VERSION 1 – AUTHOR RESPONSE

Reviewer 1

Reviewer Name: Lucy Thompson

Institution and Country: University of Glasgow, UK

Please state any competing interests or state 'None declared': None declared

Congratulations on a very good quality paper assessing an important question using good quality cohort data in two different countries. You have acknowledged the limitations of the datasets, especially in terms of differential attrition, and have taken steps to mitigate these inasmuch as is possible. The stark difference between the UK and Denmark is somewhat depressing and reflects the social differences between the two nations - a paper like this hopefully contributes to the relevant socio-political agendas. I am not a statistical specialist, so I don't feel I can comment on the analysis in any detail. However, as a non-specialist I was able to follow what you had done and why, and found the interpretation to be appropriate.

Response:

Thank you – we tried to keep the statistical detail as accessible as possible in the main document and relegate the rest to the appendices.

A small point - there was one sentence that I didn't think quite made sense. On p23 (line 29-33) you say 'Finally, it is possible that children with psychologically distressed mothers might report their children's mental health more negatively' would it not be better to say 'Finally, it is possible that mothers who are mentally distressed might report their children's mental

health more negatively'?
Overall, an excellent paper - I I

Response:
Thank you, we have changed this as suggested.

From:
"Finally, it is possible that children with psychologically distressed mothers might report their children's mental health more negatively [1]."

To:
"Finally, it is possible that children with mentally distressed mothers might report their children's mental health more negatively [1]."

Reviewer 2

Reviewer Name: Apolinaras Zaborskis
Institution and Country: Lithuanian University of Health Sciences, Lithuania
Please state any competing interests or state 'None declared': None declared

Many thanks for giving me the opportunity to review this manuscript entitled "Pathways to inequalities in child mental health – Evidence from two national birth cohorts in the UK and Denmark".

Using national birth cohort data from two countries (UK and Denmark), the authors analysed the pathways of social inequalities in child mental health problems (MHP) and how they are mediated by perinatal factors, childhood illness and maternal mental health.

The authors hypothesised that children growing up in more disadvantaged socio-economic conditions (e.g. with lower mother education) are at increased risk of MHPs due to increased exposure to perinatal, maternal and child level risk factors. They also hypothesised that these pathways may differ between countries. The findings from this study may be useful to readers who interested not only in children's mental health but also in statistical analysis when it is necessary to assess the complex effects of mediator factors on outcomes.

The article is written in a typical format. The overall rationale, design and objectives of the data analysis conducted in this study are presented clearly and concisely. There is an excellent description of the study design, sampling frame and data collection methodology for the dataset for this study. Statistical analysis was conducted using modern techniques and methodologies. A detailed supplementary appendix explains the calculations performed.

Response:
Thank you.
Since I am not a researcher in this field, I have no substantive comments on the content of this article, except some minor points that might be taken in account for revision of the manuscript or can be considered just as questions.

1. There are many tables in the article, but the data in them have little comment. This is especially true for supplementary tables.

Response:
Thank you. As suggested, we have now added a brief comment on the main point of

each table in the appendix. For example:
Appendix Table 1:

“In this table we present the results after repeating the main analysis on the risk difference scale using logistic regression with identity link function. This supplements the main analysis presented in the paper which is on the relative scale (Table 3). The results corroborate those of our main analysis.”

2. Table 3: How was estimated PM? Clearly, the formula presented on page 13 is not appropriate for this assessment.

Response:

Thank you for your question. We used the formula presented on p13 derived by Vanderweele [2]. Here is a worked example from table 3 to illustrate.

Let:

nde = 3.15

nie = 1.33

$(nde*(nie-1))/(nde*nie - 1)$

= 0.3259132

[Note estimate differs slightly in the table since this is derived from multiple imputation procedure]

3. Is it appropriate to use path analysis for the causal mediation analysis?

Response:

Thank you for the question. Yes, indeed path analyses can be used for causal mediation analysis [3]. However, we did not use path analysis in this analysis. Instead we have made use of natural effects models using the counterfactual framework in our paper. The advantages of this are that the indirect effect via multiple mediators could be simultaneously assessed such that multiple mediators could be treated as a single joint mediator, and that the natural direct and indirect effects obtained will be provided with formal causal definition as outlined in our supplementary appendix.

Thank you for considering my opinion.

Response: Thanks for the helpful feedback.

FORMATTING AMENDMENTS (if any)

Required amendments will be listed here; please include these changes in your revised version:

- Please ensure to cite all supplementary tables and figures in ascending order.

Response:

We have amended throughout the manuscript.

- Required format for figures 1 to 3

Figures can be supplied in TIFF, JPG or PDF format (figures in document, excel or powerpoint format will not be accepted), we also request that they have a resolution of at least 300 dpi and 90mm x 90mm of width.

Response:

We have amended the picture format.

References:

- 1 Najman JM, Williams GM, Nikles J, et al. Bias influencing maternal reports of child behaviour and emotional state. *Social psychiatry and psychiatric epidemiology* 2001;36:186-94.
- 2 VanderWeele T. *Explanation in causal inference: methods for mediation and interaction*: Oxford University Press 2015.
- 3 Richiardi L, Bellocco R, Zugna D. Mediation analysis in epidemiology: methods, interpretation and bias. *International journal of epidemiology* 2013;42:1511-9.

VERSION 2 – REVIEW

REVIEWER	Apolinaras Zaborskis Lithuanian University of Health Sciences
REVIEW RETURNED	14-Aug-2020
GENERAL COMMENTS	The edits to the article were done well. Thank you for answering the questions that interest me.